# β spectrin-dependent and domain specific mechanisms for Na$^+$ channel clustering

**Cheng-Hsin Liu[1,2], Ryan Seo[3], Tammy Szu-Yu Ho[1], Michael Stankewich[4], Peter J Mohler[5], Thomas J Hund[6], Jeffrey L Noebels[1,3], Matthew N Rasband[1,2]\***

[1]Department of Neuroscience, Baylor College of Medicine, Houston, United States; [2]Program in Developmental Biology, Baylor College of Medicine, Houston, United States; [3]Department of Neurology, Baylor College of Medicine, Houston, United States; [4]Department of Pathology, Yale University, New Haven, United States; [5]Department of Physiology and Cell Biology, The Ohio State University, Columbus, United States; [6]Department of Biomedical Engineering, The Ohio State University, Columbus, United States

**Abstract** Previously, we showed that a hierarchy of spectrin cytoskeletal proteins maintains nodal Na$^+$ channels (Liu et al., 2020). Here, using mice lacking β1, β4, or β1/β4 spectrins, we show this hierarchy does not function at axon initial segments (AIS). Although β1 spectrin, together with AnkyrinR (AnkR), compensates for loss of nodal β4 spectrin, it cannot compensate at AIS. We show AnkR lacks the domain necessary for AIS localization. Whereas loss of β4 spectrin causes motor impairment and disrupts AIS, loss of β1 spectrin has no discernable effect on central nervous system structure or function. However, mice lacking both neuronal β1 and β4 spectrin show exacerbated nervous system dysfunction compared to mice lacking β1 or β4 spectrin alone, including profound disruption of AIS Na$^+$ channel clustering, progressive loss of nodal Na$^+$ channels, and seizures. These results further define the important role of AIS and nodal spectrins for nervous system function.

**\*For correspondence:**
rasband@bcm.edu

**Competing interests:** The authors declare that no competing interests exist.

## Introduction

Clustered ion channels at axon initial segments (AIS) and nodes of Ranvier are essential for proper nervous system function. Spectrin tetramers, consisting of α2 and β4 spectrins have been proposed to participate in Na$^+$ channel clustering at the AIS and nodes of Ranvier. Mice lacking α2 spectrin have disrupted AIS and node integrity, and axon degeneration (*Huang et al., 2017b*; *Huang et al., 2017a*), suggesting spectrin cytoskeletons are required in these domains. Mice with mutant forms of β4 spectrin also show disrupted AIS integrity, but they have a milder phenotype than α2 spectrin-deficient mice (*Komada and Soriano, 2002*; *Lacas-Gervais et al., 2004*; *Yang et al., 2004*). Differences may reflect truncated remnants of β4 spectrin that partially execute spectrin's function. Furthermore, compensation by other β spectrins has been found at nodes of Ranvier (*Ho et al., 2014*), but has not been elucidated at the AIS.

Using conditional knockout mice, we previously showed that loss of β4 spectrin from nodes of Ranvier can be compensated for by β1 spectrin (*Liu et al., 2020*). Furthermore, we found that nodal spectrins are not required for node assembly, but rather function to maintain the molecular organization of nodes including high densities of voltage-gated Na$^+$ (Nav) channels. However, these experiments only examined nodes in dorsal root ganglion sensory neurons in order to separate the function of nodal spectrins from AIS spectrins; DRG neurons do not have an AIS (*Gumy et al., 2017*). Consistent with these findings in mice, human pathogenic variants of β4 spectrin do not have sensory neuron dysfunction (*Wang et al., 2018*). However, these individuals have severe motor axonal neuropathy. These differences suggest that an AIS may render neurons particularly susceptible

to loss of spectrin cytoskeletons. To determine why AIS are more sensitive to loss of AIS spectrins and to gain insight into the pathomechanisms of human spectrinopathies, we generated β1, β4, and β1/β4 spectrin-deficient mice.

## Results

### Loss of β4 spectrin impairs AIS nav channel clustering

To disrupt the function of β4 spectrin at AIS we generated *Nes-Cre;Sptbn4^{F/F}* mice. Mice lacking β4 spectrin show a pronounced tremor and perform significantly worse on the rotarod (*Figure 1A*) compared to littermate controls. Immunostaining showed that in the absence of β4 spectrin, AIS Nav channels and AnkG are significantly reduced in β4 spectrin-deficient mice (*Figure 1B,C*). These results are similar to those obtained in other whole-body β4 spectrin mutant mice (*Komada and Soriano, 2002*; *Lacas-Gervais et al., 2004*). Remarkably, despite the loss of β4 spectrin, super-resolution stimulated emission depletion (STED) microscopy of AIS AnkG (*Figure 1D*) showed that AnkG retained its appropriate periodic spacing (*Figure 1E*). In contrast to AIS, nodes of Ranvier in *Nes-Cre;Sptbn4^{F/F}* mice still had appropriately clustered nodal Nav channels (*Figure 1F*) since β1 spectrin and AnkR compensate for loss of β4 spectrin and AnkG at CNS nodes in the *corpus callosum* (*Figure 1G,H*; *Ho et al., 2014*; *Liu et al., 2020*). Thus, AIS are disrupted without β4 spectrin. In contrast, *Nes-Cre;Sptbn4^{F/F}* CNS nodes of Ranvier are rescued by β1 spectrin and AnkR.

### β1 spectrin is found at the AIS of some β4 spectrin-deficient neurons

β1 spectrin is expressed in motor and sensory cortex (layers II-V) at very high levels in some neurons (β1-high), but at low levels in most neurons (β1-low). Among the β1-high neurons, most are Parvalbumin-positive (PV(+)) interneurons where β1 spectrin colocalizes with AnkR (*Figure 2A–C*). In control *Sptbn4^{F/F}* mice we did not detect AIS β1 spectrin in β1-low neurons, and only very rarely was it found at the AIS of β1-high neurons (*Figure 2D,E*). Remarkably, in *Nes-Cre;Sptbn4^{F/F}* mice we found increased levels of AIS β1 spectrin in the majority of β1-high neurons (where somatic expression remained high; *Figure 2D*, arrow, 2E), but not at the AIS of β1-low neurons (*Figure 2D*, arrowheads, 2E). Despite the presence of AIS β1 spectrin in β1-high neurons, AIS Nav channels were still significantly reduced (*Figure 2F*) and the density of AIS Nav channels was comparable between β1-high and β1-low neurons (*Figure 2G*).

### Loss of β1 spectrin does not affect AIS structure or function

β1 spectrin is expressed at high levels in PV+ cells (*Figure 2A–C*). However, the normal function of neuronal β1 spectrin is unknown. To determine if β1 spectrin plays important roles in the nervous system we constructed *Nes-Cre;Sptb^{F/F}* mice. β1 spectrin-deficient mice had no discernable behavioral abnormalities and performed well on the rotarod (*Figure 2—figure supplement 1A*). β1 spectrin-deficient mice also had normal AIS (*Figure 2—figure supplement 1B*), with appropriate levels of AnkG, Nav channels, and β4 spectrin in PV(+) and PV(-) neurons (*Figure 2—figure supplement 1C–F*). Thus, like at nodes of Ranvier (*Liu et al., 2020*), β1 spectrin appears not to play critical roles in AIS structure or function when β4 spectrin is present.

### AnkR cannot compensate for AnkG at AIS

How can β1 spectrin compensate for loss of β4 spectrin at nodes, but not AIS? Since nodal Na^+ channel clustering can be rescued in β4 spectrin-deficient mice by β1 spectrin and AnkR (*Figure 1F–H*; *Liu et al., 2020*), we first determined if AnkR is found at cortical neuron AIS. We found that AnkR is not located at AIS of control *Sptbn4^{F/F}* cortical neurons despite its robust expression in β1-high neurons (*Figures 2C* and *3A*, arrows, 3B). Furthermore, but in contrast to *Nes-Cre;Sptbn4^{F/F}* nodes of Ranvier, we found no AIS AnkR in *Nes-Cre;Sptbn4^{F/F}* mice; even cortical neurons expressing high levels of AnkR did not have AnkR at the AIS (*Figure 3B*). Thus, AIS β1 spectrin cannot compensate for loss of β4 spectrin since it lacks its preferential binding partner AnkR (*Ho et al., 2014*).

Why does AnkR fail to be recruited to the AIS of *Nes-Cre;Sptbn4^{F/F}* mice when it can be recruited to nodes of Ranvier? To address this question we examined the AIS targeting of GFP-tagged AnkG

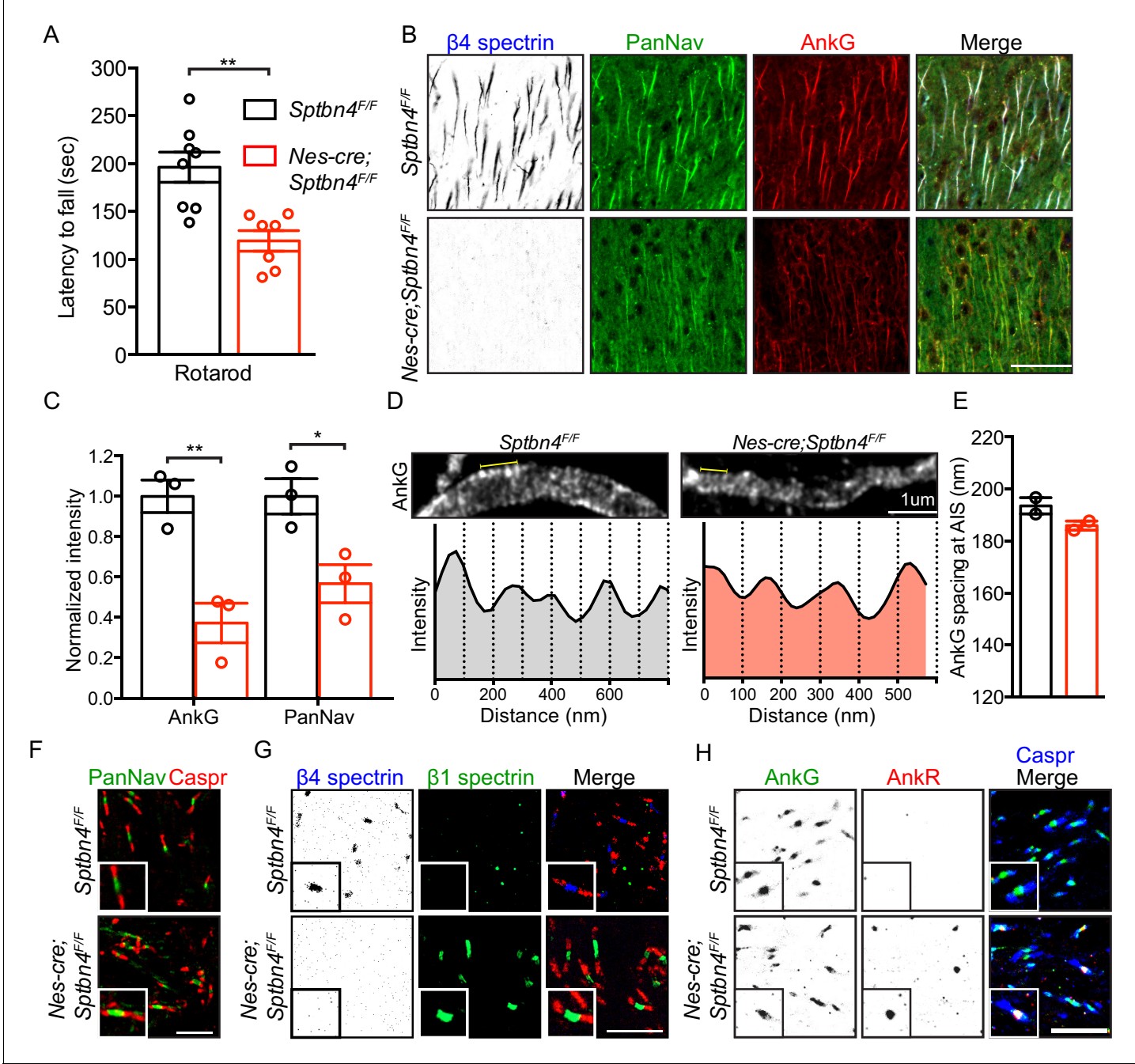

**Figure 1.** Mice lacking β4 spectrin in the central nervous system have impaired motor behavior and disrupted AIS, but intact nodal Nav clustering. (**A**) Accelerating rotarod test performed on 3 month-old $Sptbn4^{F/F}$ and $Nes-cre;Sptbn4^{F/F}$ mice. $Sptbn4^{F/F}$, N = 8; $Nes-cre;Sptbn4^{F/F}$, N = 7. Data are mean ± SEM, **p=0.0017. (**B**) Immunostaining of cortical brain sections from 3 month-old $Sptbn4^{F/F}$ and $Nes-cre;Sptbn4^{F/F}$ mice using antibodies against the β4 spectrin (blue), PanNav channels (green), and AnkG (red). Scale bar, 50 μm. (**C**) Normalized fluorescence intensity of AnkG and PanNav channel at AIS were measured from cortical brain sections of 3 month-old $Sptbn4^{F/F}$ and $Nes-cre;Sptbn4^{F/F}$ mice. N = 3 mice with a total of 51–70 AIS were measured in each genotype. Data are mean ± SEM, For AnkG, **p=0.0079; PanNav channel, *p=0.0282. (**D**) STED images of AnkG at cortical AIS from 3 month-old $Sptbn4^{F/F}$ and $Nes-cre;Sptbn4^{F/F}$ mice. The regions between the yellow lines (as shown in images) were used to generate intensity profile as shown in lower panels. (**E**) Measurements of cortical AIS AnkG spacing by STED imaging from 3 month-old $Sptbn4^{F/F}$ and $Nes-cre;Sptbn4^{F/F}$ mice. Data are mean ± SEM. $Sptbn4^{F/F}$, n = 107 spacings; $Nes-cre;Sptbn4^{F/F}$, n = 96 spacings were measured from 2 mice of each genotype. (**F**) Immunostaining of corpus callosum from 3 month-old $Sptbn4^{F/F}$ and $Nes-cre;Sptbn4^{F/F}$ mice using antibodies against PanNav channel (green), and Caspr (red). Scale bar, 10 μm. (**G**) Immunostaining of corpus callosum from 3 month-old $Sptbn4^{F/F}$ and $Nes-cre;Sptbn4^{F/F}$ mice using antibodies against β4 spectrin (blue), β1 spectrin (green), and Caspr (red). Scale bar, 10 μm. (**H**) Immunostaining of corpus callosum from 3 month-old $Sptbn4^{F/F}$ and $Nes-cre;Sptbn4^{F/F}$ mice using antibodies against Caspr (blue), AnkG (green), and AnkR (red). Scale bar, 10 μm.

*Figure 1 continued on next page*

*Figure 1 continued*

The online version of this article includes the following source data for figure 1:

**Source data 1.** Source data for *Figure 1*.

and AnkR in cultured hippocampal neurons. Ankyrin proteins consist of a membrane binding domain (MBD), a spectrin-binding domain (SBD), and a regulatory domain (RD) (*Figure 3C*). AnkG also contains a giant exon (GE) that can result in 270 and 480 kDa splice variants (*Jenkins et al., 2015*). Transfection of AnkG-270kDa-GFP or AnkR-GFP into hippocampal neurons revealed that only AnkG-270kDa-GFP is targeted to the AIS (*Figure 3D,E*). However, introduction of the GE into AnkR (AnkR/G chimera-GFP) was sufficient for it to be targeted to the AIS (*Figure 3D,E*). Thus, AnkR is not found at the AIS of *Nes-Cre;Sptbn4^{F/F}* mice since AnkR lacks the critical GE domain necessary for AIS localization.

## Loss of β1 and β4 spectrin disrupts AIS and nodal nav channel clustering and causes seizures

Human pathogenic variants of β4 spectrin cause severe neurologic dysfunction (*Knierim et al., 2017*; *Wang et al., 2018*). However, DRG sensory neuron physiology is intact, most likely due to partial compensation by β1 spectrin. To further determine if β1 spectrin contributes to nervous system function in the context of β4 spectrin-deficient neurons, we generated mice lacking both β1 and β4 spectrin. We found that *Nes-Cre;Sptb^{F/F};Sptbn4^{F/F}* mice performed worse on the rotarod than *Nes-Cre;Sptb^{F/F}*, *Nes-Cre;Sptbn4^{F/F}*, or *Sptb^{F/F};Sptbn4^{F/F}* (p<0.0001; *Figures 1A* and *4A*, *Figure 2—figure supplement 1A*, and *Video 1*). We next performed EEG recordings on 4 *Nes-Cre;Sptb^{F/F};Sptbn4^{F/F}* and 3 *Sptb^{F/F};Sptbn4^{F/F}* mice aged 1.5–3 months over prolonged periods (15–121 hr). Control mice displayed rare to infrequent spike activity but no evidence of seizures (*Figure 4B*). In β1/β4 spectrin-deficient mice, we detected very frequent interictal cortical spike discharges (299–792/hour; *Figure 4C*), and frequent spontaneous generalized spike-wave seizures (49–154/hour; *Figure 4D*) in both male and female mice. These brief (0.5–2 s) stereotyped seizure discharges occurred only during behavioral arrest. Immunostaining for AnkG and PanNav showed profound disruption of cortical AIS (*Figure 4E–F*). However, we did not observe increased neuronal cell death or neurodegeneration as indicated by antibodies against active caspase-3 or βAPP (not shown). Analysis of nodes of Ranvier in the *corpus callosum* showed that as in peripheral sensory neurons, the number of nodes with Nav channels gradually decreased over time (*Figure 4G–H*). Thus, the worsened behavioral and functional phenotypes in β1/β4 spectrin-deficient mice reflect disruption of both AIS and nodes of Ranvier and show the importance of these excitable domains for proper nervous system function.

## Discussion

β spectrins, together with α2 spectrin, form a periodic cytoskeleton in axons that is thought to confer flexibility and to protect axons from mechanical injury (*Xu et al., 2013*). In support of this idea, mouse sensory neurons lacking α2 spectrin degenerate (*Huang et al., 2017a*), mice lacking CNS β2 spectrin have extensive axon degeneration (*Lorenzo et al., 2019*), and loss of β3 spectrin causes cerebellar Purkinje neuron degeneration and spinocerebellar ataxia type 5 (SCA5) (*Ikeda et al., 2006*). Interestingly, we found no CNS pathology in β1 spectrin-deficient mice, suggesting that this spectrin is dispensable. β spectrins may also be restricted to or enriched at precise subcellular domains like AIS, paranodal junctions, and nodes of Ranvier, suggesting that their roles extend beyond protecting axons from mechanical injury (*Liu and Rasband, 2019*). Consistent with this idea, mice lacking β2 spectrin in sensory neuron axons do not degenerate, but instead have Kv1 K+ channels that move into paranodal regions formerly occupied by β2 spectrin (*Zhang et al., 2013*). β spectrins can also serve as a signaling platform. In cardiomyocytes, β4 spectrin regulates membrane excitability through coordinating CAMK2-mediated modification to Nav 1.5 channels for proper subcellular localization (*Hund et al., 2010*). This β4 spectrin-mediated signaling platform is dissociated under stress and further induces protective gene expression by STAT3 (*Unudurthi et al., 2018*). It will be interesting to determine whether β4 spectrin plays similar roles in the nervous system. For

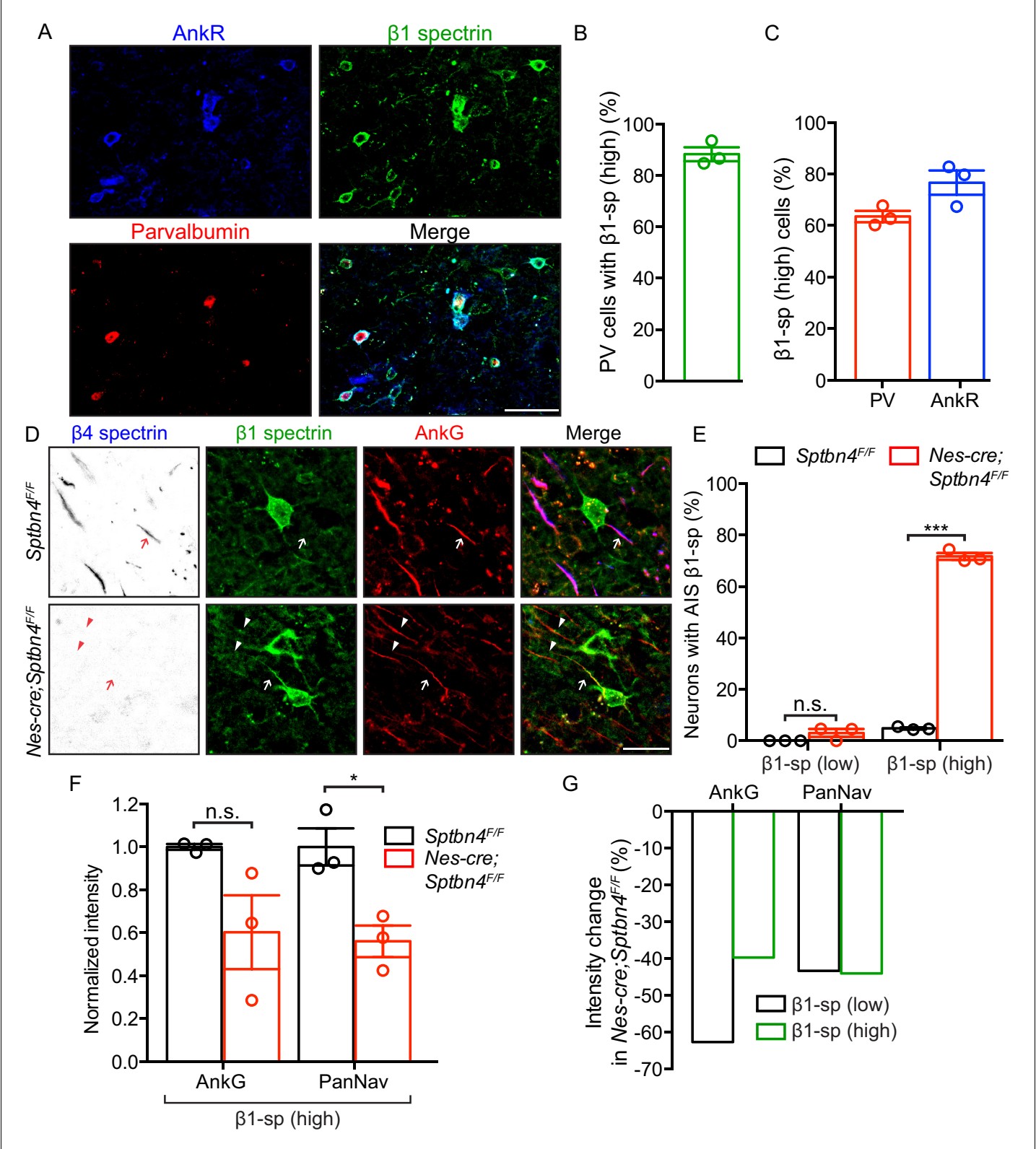

**Figure 2.** β1 spectrin is localized at the AIS of parvalbumin-positive neurons in β4 spectrin deficient mice. (**A**) Immunostaining of brain cortical sections from 3 month-old *Sptb^F/F* mice using antibodies against the AnkR (blue), β1 spectrin (green), and parvalbumin (PV, red). Scale bar, 50 μm. (**B**) The percentage of PV-positive neurons labeled with high β1 spectrin in 3 month-old *Sptb^F/F* mice cortex. N = 3 animals, with total 166 neurons counted. (**C**) The percentage of high β1 spectrin signal in cortical neurons labeled with PV or AnkR in 3 month-old *Sptb^F/F* mice. N = 3 animals, with total 231 and

*Figure 2 continued on next page*

*Figure 2 continued*

251 neurons counted, respectively. (**D**) Immunostaining of brain cortical sections from 3 month-old *Sptbn4^{F/F}* and *Nes-cre;Sptbn4^{F/F}* mice using antibodies against β4 spectrin (blue), β1 spectrin (green), and AnkG (red). AIS of high and low β1 spectrin expression neurons are indicated by arrows and arrowheads, respectively. Scale bar, 25 μm. (**E**) The percentage of neurons with AIS β1 spectrin in β1 spectrin low and β1 spectrin high 3 month-old *Sptbn4^{F/F}* and *Nes-cre;Sptbn4^{F/F}* mice. N = 3 mice in each genotype. For *Sptbn4^{F/F}* mice, 122 and 125 β1 spectrin low/high neurons were counted, respectively; for *Nes-cre; Sptbn4^{F/F}* mice, 122 and 120 β1 spectrin low/high neurons were counted, respectively. For β1 spectrin (low) population, p=0.1162; for β1 spectrin (high) population, \*\*\*p=1.14E-06. (**F**) Normalized fluorescence intensity of AnkG and PanNav channel at AIS from β1 spectrin (high) cortical neurons measured in 3 month-old *Sptbn4^{F/F}* and *Nes-cre;Sptbn4^{F/F}* mice. N = 3 mice with a total of 50–63 AIS were measured in each neuron type per genotype. Data are mean ± SEM, For AnkG, p=0.0829; PanNav channel, \*p=0.018. (**G**) The average changes of percentage of fluorescence intensity of AnkG and PanNav channel in β1 spectrin low and β1 spectrin high neurons in 3 month-old *Sptbn4^{F/F}* and *Nes-cre;Sptbn4^{F/F}* mice.

The online version of this article includes the following source data and figure supplement(s) for figure 2:

**Source data 1.** Source data for *Figure 2*.
**Figure supplement 1.** Mice lacking β1 spectrin in the central nervous system show normal motor performance and AIS structure.
**Figure supplement 1—source data 1.** Source data for *Figure 2—figure supplement 1*.

example, β4 spectrin-mediated signaling may influence downstream transcriptional regulation for axon regeneration after spectrin proteolysis by calpains.

Previous efforts to define the function of β4 spectrin in the nervous system relied on mice with mutations resulting in truncated forms of the protein (*Komada and Soriano, 2002*; *Yang et al., 2004*). Although these studies reported disrupted AIS and nodes of Ranvier, they did not examine possible contributions from other β spectrins (e.g. β1 spectrin). In addition, these studies could not determine if nodes were affected because of a disrupted AIS. In this work, the availability of *Sptb^{F/F}* and *Sptbn4^{F/F}* mice made it possible for us to further define the functions of β1 and β4 spectrin.

At nodes, β1 and β4 spectrin function to stabilize and maintain AnkG and Nav channels, and loss of these nodal spectrins leads to the eventual reduction in clustered Nav channels, and axonal injury (*Liu et al., 2020*). Here, we show that loss of β4 spectrin and the failure of β1 spectrin to compensate for its loss, causes AIS proteins to be far less stable than nodal proteins. This observation helps explain the phenotypes of human pathogenic *SPTBN4* variants (*Wang et al., 2018*). We speculate that the stability of nodal proteins compared to AIS proteins, even in the absence of β1 and β4 spectrin, depends on the flanking paranodal domains where the myelin sheath attaches to the axon. These paranodal axonal domains are enriched with an α2/β2 spectrin cytoskeleton that functions as an independent mechanism to cluster AnkG, β4 spectrin, and Nav channels in CNS axons (*Amor et al., 2017*). Thus, nodal proteins may be stabilized and maintained in the plasma membrane by both nodal and paranodal cytoskeletons, while AIS proteins rely mainly on β4 spectrin. One recent report indicated that β2 spectrin may also be found at AIS of hippocampal neurons in culture (*Lazarov et al., 2018*). However, even if β2 spectrin is present at AIS in vivo, it cannot compensate for loss of β4 spectrin.

Our experiments revealed that β1 spectrin cannot compensate for AIS β4 spectrin, even in PV(+) cells expressing very high levels of AnkR and β1 spectrin. This is because β1 spectrin is the preferential binding partner for AnkR (*Ho et al., 2014*), and AnkR cannot be recruited to AIS. AIS require AnkG for their assembly and to cluster AIS Nav channels (*Zhou et al., 1998*); AnkG also recruits β4 spectrin to AIS (*Yang et al., 2007*). The enrichment of AnkG at AIS depends on the domain encoded by its giant exon (*Jenkins et al., 2015*). Our studies support this conclusion since AnkR/AnkG chimeras can be recruited to the AIS. AnkG's giant exon participates in interactions with β4 spectrin (*Jenkins et al., 2015*), EB1/3 (*Leterrier et al., 2011*) and NDEL1 (*Kuijpers et al., 2016*). Together with these proteins, giant AnkG stabilizes the AIS cytoskeleton and regulates polarized trafficking. Giant AnkG also interacts with GABARAP to stabilize GABA receptors in the somatodendritic domain of neurons (*Nelson et al., 2018*).

In summary, our results show that although AIS are thought to be the evolutionary precursors to nodes of Ranvier (*Hill et al., 2008*), they lack the molecular flexibility of nodes in that they require AnkG and β4 spectrin, while nodes can assemble from both AnkG/β4 spectrin and AnkR/β1 spectrin.

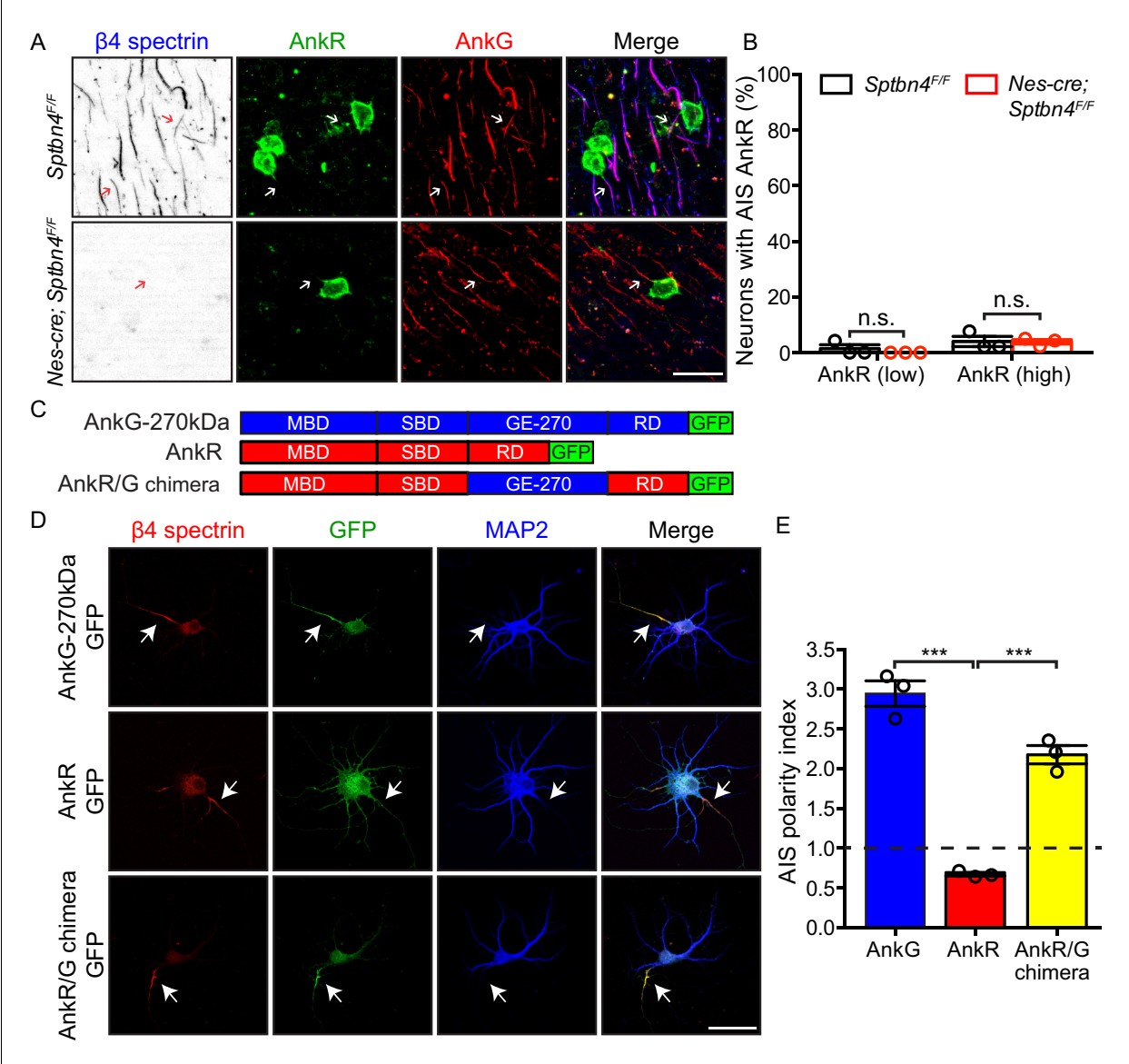

**Figure 3.** AnkyrinR fails to target to the AIS. (**A**) Immunostaining of brain cortical sections from 3 month-old *Sptbn4$^{F/F}$* and *Nes-cre;Sptbn4$^{F/F}$* mice using antibodies against β4 spectrin (blue), AnkR (green), and AnkG (red). AIS of high AnkR expression neurons are indicated by arrows. Scale bar, 25 μm. (**B**) The percentage of neurons with AnkR in the AIS in two subpopulations (AnkR low/high) of 3 month-old *Sptbn4$^{F/F}$* and *Nes-cre;Sptbn4$^{F/F}$* mice. N = 3 mice in each genotype. For *Sptbn4$^{F/F}$* mice, 150 and 129 AnkR low/high neurons were counted; for *Nes-cre; Sptbn4$^{F/F}$* mice, 127 and 128 AnkR low/high neurons were counted. For AnkR (low) population, p=0.3739; for AnkR (high), p=0.9497. (**C**) Domain structure and design of AnkG-270kDa-GFP, AnkR-GFP, and AnkR/G chimera-GFP expression constructs. (**D**) Immunostaining of cultured rat hippocampal neurons at DIV10 after transfected with AnkG-270kDa-GFP, AnkR-GFP, or AnkR/G chimera-GFP expression plasmids. Antibodies were used against β4 spectrin (red), GFP (green), and the somatodendritic marker MAP2 (blue). AIS are indicated by arrows. Scale bar, 50 μm. (**E**) Quantification of the ratio of GFP signal intensity at AIS versus proximal dendrite in AnkG-270kDa-GFP, AnkR-GFP, or AnkR/G chimera-GFP transfected cultured hippocampal neurons. N = 3 batches of cultured neurons for each transfected plasmids, with total 43–49 neurons were measured for each plasmid. Data are mean ± SEM. For AnkG-270kDa-GFP versus AnkR-GFP, ***p=0.0002; for AnkR-GFP versus AnkR/G chimera-GFP, ***p=0.0002.

The online version of this article includes the following source data for figure 3:

**Source data 1.** Source data for *Figure 3*.

We speculate that this difference is a consequence of myelin. Specifically, the difference between the intrinsic mechanisms of AIS assembly (*Galiano et al., 2012*) and the multiple glia-dependent (extrinsic) mechanisms of node assembly.

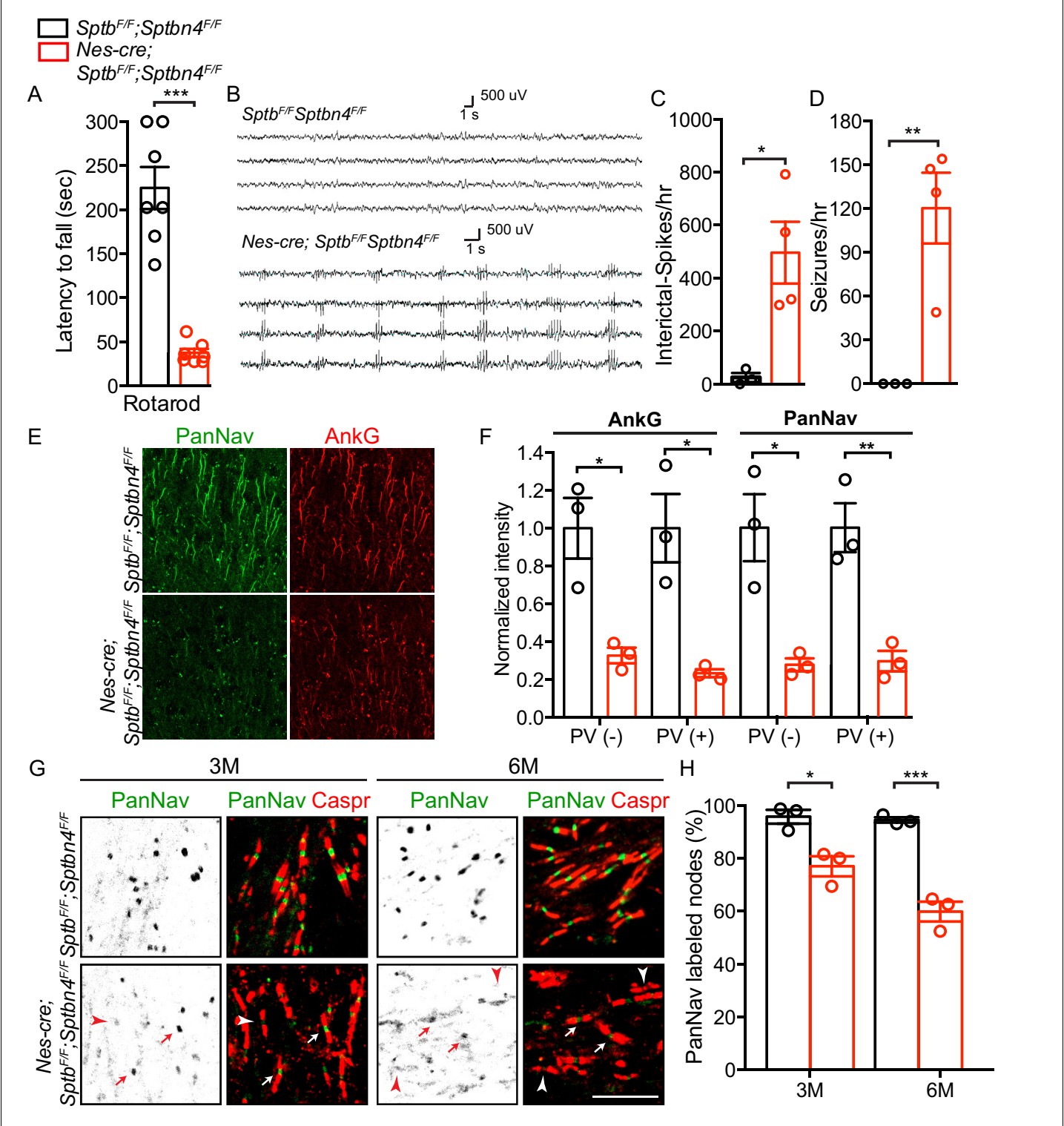

**Figure 4.** Mice lacking both β1 and β4 spectrin have severe motor impairment, epileptic activity, disrupted AIS, and gradual loss of nodal Nav channel clustering. (**A**) Accelerating rotarod test performed on 3 month-old *Sptb*$^{F/F}$*; Sptbn4*$^{F/F}$ and *Nes-cre;Sptb*$^{F/F}$*; Sptbn4*$^{F/F}$ mice, with N = 7 animals tested per genotype. Data are mean ± SEM, ***p=5.78E-06. (**B**) Video EEG monitoring of awake and behaving 3 month-old mice showed generalized seizure discharges in *Nes-cre;Sptb*$^{F/F}$*; Sptbn4*$^{F/F}$ mice that were not detected in *Sptb*$^{F/F}$*; Sptbn4*$^{F/F}$ littermates. (**C–D**) Quantification of interictal-spikes/hr and seizures/hr in *Sptb*$^{F/F}$*; Sptbn4*$^{F/F}$ and *Nes-cre;Sptb*$^{F/F}$*; Sptbn4*$^{F/F}$ mice. N = 3 and 4 for *Sptb*$^{F/F}$*; Sptbn4*$^{F/F}$ and *Nes-cre;Sptb*$^{F/F}$*; Sptbn4*$^{F/F}$ mice, respectively. Data are mean ± SEM. For interictal-spikes/hr, *p=0.0194; for seizures/hr, **p=0.0085. (**E**) Immunostaining of brain cortical sections from 3 month-old *Sptb*$^{F/F}$*; Sptbn4*$^{F/F}$ and *Nes-cre;Sptb*$^{F/F}$*; Sptbn4*$^{F/F}$ mice using PanNav (green) and AnkG (red) antibodies. Scale bar, 50 μm. (**F**) Normalized

*Figure 4 continued on next page*

*Figure 4 continued*

fluorescence intensity for AnkG and PanNav at AIS in PV(-) and PV(+) neurons in cortex from 3 month old *Sptb^F/F^; Sptbn4^F/F^* and *Nes-cre;Sptb^F/F^; Sptbn4^F/F^* mice. N = 3 mice with a total of 37–50 AIS measured in each neuron type per genotype. Data are mean ± SEM. For AnkG in PV(-) and PV(+) neurons, *p=0.0152 and *p=0.0135, respectively; for PanNav in PV(-) and PV(+) neurons, *p=0.0159 and **p=0.0073, respectively. (**G**) Immunostaining of corpus callosum from 3 and 6 month-old *Sptb^F/F^; Sptbn4^F/F^* and *Nes-cre;Sptb^F/F^; Sptbn4^F/F^* mice using PanNav (green) and Caspr (red) antibodies. Arrows indicate intact nodal Nav clusters, whereas arrowheads indicate nodes devoid of Nav channels. Scale bar, 10 μm. (**H**) Quantification of the percentage of corpus callosum nodes labeled for Nav channels in *Sptb^F/F^; Sptbn4^F/F^* and *Nes-cre;Sptb^F/F^; Sptbn4^F/F^* mice at the indicated ages. Data are mean ± SEM. N = 3 animals with a total of 345–377 nodes counted in each genotype per age. For 3 month-old, *p=0.0152; 6 month-old, ***p=0.0009.

The online version of this article includes the following source data for figure 4:

**Source data 1.** Source data for *Figure 4*.

# Materials and methods

## Key resources table

| Reagent type (species) or resource | Designation | Source or reference | Identifiers | Additional information |
|---|---|---|---|---|
| Gene (*Mus musculus*) | *Sptb* | https://www.ncbi.nlm.nih.gov/gene/20741 | Gene ID: 20741 | |
| Gene (*M. musculus*) | *Sptbn4* | https://www.ncbi.nlm.nih.gov/gene/80297 | Gene ID: 80297 | |
| Genetic reagent (*M. musculus*) | Nestin-cre | The Jackson Laboratory | Stock No:003771 | |
| Genetic reagent (*M. musculus*) | *Sptb^flox/flox^* | PMID:32052742 | | |
| Genetic reagent (*M. musculus*) | *Sptbn4^flox/flox^* | PMID:30226828 | | |
| Antibody | Anti-Ankyrin G (Mouse monoclonal) | Neuromab | Clone: N106/36; RRID:AB_10673030 | IF (1:500) |
| Antibody | Anti-Ankyrin G (Mouse monoclonal) | Neuromab | Clone N106/65; RRID:AB_10675130 | IF (1:100) for STED |
| Antibody | Anti-Ankyrin R (Mouse monoclonal) | Neuromab | Clone N380/A10; RRID:AB_2491109 | IF (1:500) |
| Antibody | Anti-Parvalbumin (Mouse monoclonal) | Neuromab | Clone L114/3; RRID:AB_2651167 | IF (1:500) |
| Antibody | Anti-PanNav (Mouse monoclonal) | Neuromab | Clone: N419/78; RRID:AB_2493099 | IF (1:300) |
| Antibody | Anti-PanNav (Mouse monoclonal) | Sigma-Aldrich | Clone: K58/35; RRID:AB_477552 | IF (1:300) |
| Antibody | Anti-β1 spectrin (Mouse monoclonal) | Neuromab | Clone: N385/21; RRID:AB_2315815 | IF (1:500) |
| Antibody | Anti-β4 spectrin SD antibody (Rabbit polyclonal) | PMID:28123356 | | IF (1:500) WB (1:1000) |
| Antibody | Anti-Ankyrin R (Rabbit polyclonal) | PMID:25362473 | | IF (1:500) |

*Continued on next page*

*Continued*

| Reagent type (species) or resource | Designation | Source or reference | Identifiers | Additional information |
|---|---|---|---|---|
| Antibody | Anti-Caspr (Rabbit polyclonal) | PMID:10460258 | RRID:AB_2572297 | IF (1:500) |
| Antibody | Anti-Pan Neurofascin (Chicken polyclonal) | R and D Systems | Cat.#: AF3235; RRID:AB_10890736 | IF (1:500) |
| Antibody | Anti-MAP2 (Chicken polyclonal) | Encor | Cat.#: CPCA-MAP2; RRID:AB_2138173 | IF (1:500) |
| Antibody | Anti-GFP (Rat monoclonal) | Biolegend | Cat.#: 338002; RRID:AB_1279414 | IF (1:500) |
| Antibody | Anti-Parvalbumin (Rabbit polyclonal) | Novus | RRID:AB_791498 | IF (1:500) |
| Antibody | Anti-active Caspase 3 (Rabbit polyclonal) | R and D Systems | RRID:AB_2243952 | IF (1:500) |
| Antibody | Anti-βAPP (Rabbit polyclonal) | Thermo Fisher Scientific | RRID:AB_2533902 | IF (1:1000) |
| Sequence-based reagent | Genotyping primer for $Sptbn4^{flox/flox}$ mouse (sense) | PMID:30226828 | | 5′-GAGCTGCATAAGT TCTTCAGCGATGC-3′ |
| Sequence-based reagent | Genotyping primer for $Sptbn4^{flox/flox}$ mouse (anti-sense) | PMID:30226828 | | 5′-ACCCCATCTCAAC TGGCTTTCTTGG-3′ |
| Sequence-based reagent | Genotyping primer for $Sptb^{flox/flox}$ mouse (sense) | PMID:32052742 | | 5′- ACAGAGACAGA TGGCCGAAC-3′ |
| Sequence-based reagent | Genotyping primer for $Sptb^{flox/flox}$ mouse (anti-sense) | PMID:32052742 | | 5′-CTCTGGTTCCCA GGAGAGC-3′ |
| Sequence-based reagent | Genotyping primer for *Avil-cre* mouse (primer 1) | PMID:29038243 | | 5′-CCCTGTTCACTG TGAGTAGG-3′ |
| Sequence-based reagent | Genotyping primer for *Avil-cre* mouse (primer 2) | PMID:29038243 | | 5′- AGTATCTGGTAG GTGCTTCCAG-3′ |
| Sequence-based reagent | Genotyping primer for *Avil-cre* mouse (primer 3) | PMID:29038243 | | 5′-GCGATCCCTGAA CATGTCCATC-3′ |
| Transfected construct (Rat) | pEGFP-N1-AnkG-270kDa | PMID:9744885 | | Transfected construct (Rat) |
| Transfected construct (Human) | pEGFP-N1-AnkR | This paper | | Transfected construct (Human) Rasband laboratory |
| Transfected construct (Human/Rat) | pEGFP-N1-AnkR/G chimera | This paper | | Transfected construct (Human/Rat) Rasband laboratory |
| Sequence-based reagent | AnkR-F | This paper | | 5′-ATCTCGAGATGCCC TATTCTGTGG-3′ Rasband laboratory |
| Sequence-based reagent | AnkR-R | This paper | | 5′-AGCTTGAGGGGG TTGGGTGTCGA-3′ Rasband laboratory |
| Sequence-based reagent | pEGFP-N1-F | This paper | | 5′-CCAACCCCCTCAA GCTTCGAATTCTG-3′ Rasband laboratory |

*Continued on next page*

*Continued*

| Reagent type (species) or resource | Designation | Source or reference | Identifiers | Additional information |
|---|---|---|---|---|
| Sequence-based reagent | pEGFP-N1-R | This paper | | 5'-TAGGGCATCTCG AGATCTGAGTCC-3' Rasband laboratory |
| Sequence-based reagent | AnkR-SBD-F | This paper | | 5'-CCCCTGGTACAGG CAACGTTCCCGGA GAATG-3' Rasband laboratory |
| Sequence-based reagent | AnkR-SBD-R | This paper | | 5'-ACTGTTTTGTAT CGCAGGGCCAG-3' Rasband laboratory |
| Sequence-based reagent | AnkG-RD-F | This paper | | 5'-TGCGATACAAAA CAGTTGAACGGAG-3' Rasband laboratory |
| Sequence-based reagent | AnkG-RD-R | This paper | | 5'- GTACCGTCGACTGCA GAATTCGGTGGGC TTTCTTCTC-3' Rasband laboratory |
| Sequence-based reagent | AnkR-RD-F | This paper | | 5'- TCCGATATCAG CATTCTCAGTG AGTCC-3' Rasband laboratory |
| Sequence-based reagent | AnkR-RD-R | This paper | | 5'- TAGAATTCGGGG GTTGGGTGTCGAGGTG-3' Rasband laboratory |
| Commercial assay or kit | GeneArt Seamless Cloning and Assembly Kit | Thermo Fisher Scientific | Cat#: A13288 | |
| Software, algorithm | Zen | Carl Zeiss | RRID:SCR_013672 | |
| Software, algorithm | Labchart 8.0 | ADI Systems | RRID:SCR_017551 | |
| Software, algorithm | Leica Application Suite X | Leica | RRID:SCR_013673 | |
| Software, algorithm | Fiji | National Institutes of Health | RRID:SCR_002285 | |
| Software, algorithm | Prism | Graph Pad | RRID:SCR_002798 | Version 6 |

## Animals

*Sptb*$^{F/F}$ and *Sptbn4*$^{F/F}$ mice were generated as described previously (*Unudurthi et al., 2018*; *Liu et al., 2020*). Both *Sptb*$^{F/F}$ and *Sptbn4*$^{F/F}$ mice were maintained on a mixed C57BL/6 and 129/sv background. Nestin-cre (*Nes-cre*) mice were purchased from the Jackson laboratory (Stock No:003771). Both male and female mice were used in our studies. All experiments comply with the National Institutes of Health Guide for the Care and Use of Laboratory Animals and were approved by the Baylor College of Medicine Institutional Animal Care and Use Committee.

## Antibodies

The following mouse monoclonal primary antibodies were purchased from the UC Davis/NIH Neuro-Mab facility: AnkG (106/36; RRID:AB_10673030), AnkG (106/65; RRID:AB_10675130), PanNav (N419/78; RRID:AB_2493099), β1 spectrin (N385/21; RRID:AB_2315815), Parvalbumin (L114/3; RRID: AB_2651167), AnkR (N388A/10, RRID:AB_2491109). Other antibodies were sourced as follows: mouse anti-PanNav (Sigma-Aldrich K58/35; RRID:AB_477552), chicken anti-MAP2 (Encor cat. CPCA-MAP2; RRID:AB_2138173), rat anti-GFP (Biolegend cat. 338002; RRID:AB_1279414), chicken anti-Pan-Neurofascin (R and D Systems cat. AF3235; RRID:AB_10890736), rabbit anti-βAPP (Thermo

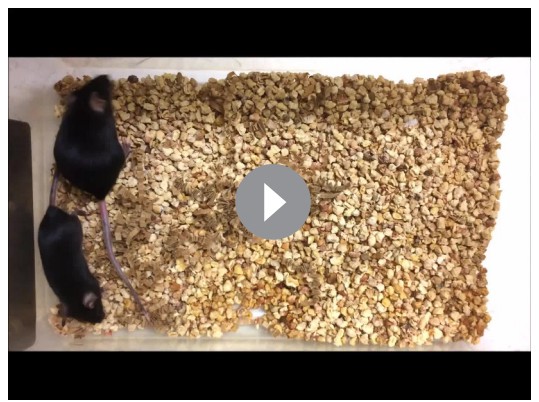

**Video 1.** 6 month-old $Sptb^{F/F}$; $Sptbn4^{F/F}$ and $Nes$-$cre$; $Sptb^{F/F}$; $Sptbn4^{F/F}$. Mice lacking β1 and β4-spectrin in the central nervous system showed motor impairments. https://elifesciences.org/articles/56629#video1

Fisher Scientific, RRID:AB_2533902), rabbit anti-Parvalbumin (Novus, RRID:AB_791498), rabbit anti-active Caspase3 (R and D Systems, RRID:AB_2243952). The following antibodies were described previously: rabbit anti-βIV Spectrin SD antibodies (*Yoshimura et al., 2016*); rabbit anti-AnkR (*Ho et al., 2014*); rabbit anti-Caspr (RRID: AB_2572297; *Rasband et al., 1999*). Secondary antibodies were purchased from Thermo Fisher Scientific and Jackson ImmunoResearch Laboratories.

## DNA constructs

The full-length rat 270 kDa AnkG-GFP has been described (*Zhang and Bennett, 1998*). The Flag-tagged full-length human AnkR (Flag-AnkR) was generated from the Ank1 isoform 1 (NM_020476.2 CDS) as described previously (*Ho et al., 2014*). The construct pEGFP-N1-AnkR was made with GeneArt Seamless Cloning and Assembly Kit (Thermo Fisher Scientific, cat. No. A13288). The primers for amplifying full-length AnkR from Flag-AnkR are forward: ATCTCGAGATGCCCTATTCTGTGG, and reverse: AGCTTGAGGGGGTTGGGTGTCGA. The primers for amplifying pEGFP-N1 are forward: CCAACCCCCTCAAGCTTCGAATTCTG, and reverse: TAGGGCATCTCGAGATCTGAGTCC. To generate pEGFP-N1-AnkR/G chimera, we first generate pEGFP-N1-AnkR (MBD-SBD)-AnkG (GE-270-RD) construct. pEGFP-N1-full length AnkR was cut with restriction enzymes EcoRI and Acl1 and ligated with two fragments, AnkR SBD C-terminal half and AnkG GE-270-RD, using GeneArt Seamless Cloning and Assembly Kit (Thermo Fisher Scientific, cat. No. A13288). The primers for amplifying AnkR SBD C-terminal half from pEGFP-N1-AnkR are forward: CCCCTGGTACAGGCAACGTTCCCGGAGAATG, and reverse: ACTGTTTTGTATCGCAGGGCCAG. The primers for amplifying AnkG GE270-RD from rat 270 kDa AnkG-GFP are forward: TGCGATACAAAACAGTTGAACGGAG, and reverse: GTACCGTCGACTGCAGAATTCGGTGGGCTTTCTTCTC. After generating pEGFP-N1-AnkR (MBD-SBD)-AnkG (GE-270-RD) construct, we constructed pEGFP-N1-AnkR (MBD-SBD)-AnkG (GE-270)-AnkR (RD) (i.e., AnkR/G chimera), by amplifying DNA fragment encoding AnkR (RD) by PCR using Flag-AnkR as template and introduced into the EcoRV-EcoRI sites of the pEGFP-N1-AnkR (MBD-SBD)-AnkG (GE-270-RD) construct. The primers for amplifying AnkR (RD) are forward: TCCGATATCAGCATTCTCAGTGAGTCC, and reverse: TAGAATTCGGGGGGTTGGGTGTCGAGGTG.

## Hippocampal neuron culture and transfection

Hippocampi were isolated and dissociated from E18.5 Sprague Dawley rat embryos. Neurons were plated on poly-D-Lysine and laminin-coated glass coverslips, and cultured in Neurobasal medium containing 1% Glutamax, 1% Penicillin/Streptomycin and 2% B27 supplement in a 5% $CO_2$ incubator. At DIV7, DNA constructs were transfected into cultured neurons using Lipofectamine 2000; after 2 days, neurons were fixed by 4% PFA and proceed to immunostaining. Above reagents were sourced from Thermo Fisher Scientific.

## Behavioral testing and electroencephalogram (EEG) recording

For accelerating rotarod test, mice were conditioned on rotating rod (Ugo Basile) with steady 5 rpm for 5 min. After 1 hr break, mice were placed on accelerated rotarod starting with 4 rpm to 40 rpm in 5 min. Latency to fall were recorded and averaged from 3 trials, which with 30 min breaks between trials. For EEG recording, mice were anesthetized with isoflurane (2.0–4% in oxygen, Patterson Veterinary Vaporizer), and silver wire electrodes (0.005″ diameter) soldered to a connector were surgically implanted bilaterally into the subdural space over frontal and parietal cortex. Mice were allowed to recover for 14 days before recording. Simultaneous video-EEG and behavioral monitoring (Labchart 8.0, ADI Systems) was performed during 24 hr sessions in adult (aged >6 weeks) mice of

either sex. EEG was recorded while mice moved freely in the test cages. All EEG signals were amplified by a g.BSAMP biosignal amplifier (Austria), digitized by PowerLab with a 0.3 Hz high-pass and 60 Hz low-pass filter (ADInstruments, Dunedin, New Zealand) and acquired via Labchart 8.0 (ADInstruments). EEGs were reviewed by two trained observers.

### Immunofluorescence and stimulated emission depletion (STED) microscopy

Procedures of mice tissue collection and preparation for immunostaining were described previously (*Liu et al., 2020*). Immunofluorescence images were captured by Axio-imager Z1 microscope or Axio-observer Z1 microscope fitted with an AxioCam digital camera, and collected by Zen software. All of these apparatus were sourced from Carl Zeiss MicroImaging. For STED microscopy, tissue sections were prepared through regular procedures except mounting using ProLong Diamond Antifade Mountant (Thermo Fisher Scientific, P36965). Imaging was performed on Leica TCS SP8X STED3x super-resolution microscope system (Leica) with 592 nm pulsed excitation laser, a pulsed 775 nm STED laser, and a 100X oil immersion objective lens (N.A. 1.4). Pixel size were around 17–30 nm among images. Deconvolution of image was performed by default LIGHTNING settings in LAS X Software (Leica). Measurements of fluorescence intensity and linear intensity profile were performed using FIJI (National Institutes of Health) and Zen (Carl Zeiss MicroImaging).

### Statistical analysis

Unpaired, two-tailed Student's t-test was performed for statistical analysis unless otherwise indicated. Data were collected and processed randomly and analyzed using GraphPad Prism and Microsoft Excel. No statistical methods or power analysis were used to predetermine sample sizes, but our sample sizes are similar to those reported previously (*Susuki et al., 2013*). Data distribution was assumed to be normal. Experimenters were blinded to genotype in the following experiments: all behavioral experiments comparing $Sptb^{F/F}$ and $Nes$-cre; $Sptb^{F/F}$ mice, all behavioral experiments comparing $Sptbn4^{F/F}$ and $Nes$-cre; $Sptbn4^{F/F}$ mice, all analyses of AnkG spacing at AIS, all analyses of AnkG, Nav channel and GFP fluorescence intensity, and all analyses of EEG recordings. Experimenters were not blinded to genotype in the behavioral tests comparing $Sptb^{F/F}$; $Sptbn4^{F/F}$ and $Nes$-cre; $Sptb^{F/F}$; $Sptbn4^{F/F}$ mice due to obvious motor impairments. No data points were excluded.

## Acknowledgements

The work reported here was supported by the following research grants: NIH NS044916 (MNR); NIH NS069688 (MNR); NIH NS29709 (JLN); and by the Dr. Miriam and Sheldon G Adelson Medical Research Foundation (MNR). We thank Dr. Dinghui Yu at the Jan and Dan Duncan Neurological Research Institute microscopy core for help with STED microscopy.

## Additional information

### Funding

| Funder | Grant reference number | Author |
|---|---|---|
| National Institutes of Health | NS044916 | Matthew N Rasband |
| National Institutes of Health | NS069688 | Matthew N Rasband |
| National Institutes of Health | NS29709 | Jeffrey L Noebels |
| Dr. Miriam and Sheldon G. Adelson Medical Research Foundation | | Matthew N Rasband |
| Mission Connect, a program of TIRR Foundation | | Matthew N Rasband |

The funders had no role in study design, data collection and interpretation, or the decision to submit the work for publication.

## Author contributions
Cheng-Hsin Liu, Conceptualization, Formal analysis, Investigation, Methodology, Writing - original draft, Writing - review and editing; Ryan Seo, Formal analysis, Investigation; Tammy Szu-Yu Ho, Investigation; Michael Stankewich, Peter J Mohler, Thomas J Hund, Resources; Jeffrey L Noebels, Formal analysis, Funding acquisition, Investigation, Writing - original draft; Matthew N Rasband, Conceptualization, Supervision, Funding acquisition, Writing - original draft, Project administration, Writing - review and editing

## Author ORCIDs
Cheng-Hsin Liu https://orcid.org/0000-0002-4582-4551
Michael Stankewich https://orcid.org/0000-0003-4472-9162
Matthew N Rasband https://orcid.org/0000-0001-8184-2477

## Ethics
Animal experimentation: This study was performed in strict accordance with the recommendations in the Guide for the Care and Use of Laboratory Animals of the National Institutes of Health. All of the animals were handled according to approved institutional animal care and use committee (IACUC) protocols (AN-4634) at Baylor College of Medicine.

## Decision letter and Author response
Decision letter https://doi.org/10.7554/eLife.56629.sa1
Author response https://doi.org/10.7554/eLife.56629.sa2

# Additional files

## Supplementary files
• Transparent reporting form

## Data availability
All data generated or analyzed during this study are included in the manuscript and supporting files. Source data files have been provided for all figures.

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
