## [Decision Letter]

**Acceptance summary:**

The current study addressed the difference in molecular mechanisms underlying the node and axon initial segment. This study further showed the requirement of β spectrins in localizing Na^+^ channel to the axon initial segment.

**Decision letter after peer review:**

Thank you for submitting your article "β spectrin-dependent and domain specific mechanisms for Na^+^ channel clustering" for consideration by *eLife*. Your article has been reviewed by two peer reviewers, and the evaluation has been overseen by Kang Shen as the Reviewing Editor and Olga Boudker as the Senior Editor The following individual involved in a review of your submission has agreed to reveal their identity: Vann Bennett (Reviewer #2).

The reviewers have discussed the reviews with one another and the Reviewing Editor has drafted this decision to help you prepare a revised submission.

Summary:

Both reviewers agree that this manuscript are of high quality and is in principle appropriate for publication as a Research Advance in *eLife*. The reviewers agree that the manuscript provides new data concerning the role of β spectrins at the axon initial segment (AIS). Previously, the authors demonstrated a hierarchy of spectrin cytoskeletal proteins in maintaining nodal Na^+^ channel clustering (Liu et al., 2020). β4 and β1 spectrins functionally compensate with one another, sequestering their appropriate binding partners – AnkG and AnkR respectively at the nodes of Ranvier. In the current manuscript by Liu et al., the authors show that this hierarchy/compensation is absent at the AIS, and that this is due to the lack of the targeting domain in AnkR to the AIS. Combining both in vivo and in vitro data, the manuscript provides compelling, high quality and rigorous data to illustrate the β4 spectrin-dependent role at the AIS, as well as the striking difference in the mechanism for Na^+^ channel clustering between nodes of Ranvier and the AIS.

Essential revisions:

The reviewers also found that there are several points that can be improved by including additional discussion.

1) The discussion of localization of β1 spectrin and AnkyrinR in a subset of neurons should note that both of these proteins are present predominantly in the cell body, both in wildtype and β4 spectrin null neurons. While some β1 staining can be detected in the β4 knockouts, this is minimal in comparison to the cell body.

2) The Introduction is a single paragraph, and could be expanded.

3) The specialized function of β4 spectrin in recruiting CamKinase2 (missing in other β spectrins) should be mentioned. Similarly, discussion of specialized roles of giant AnkyrinG (ex-interactions with GABARAP, Nudel, EB proteins, and as a signaling platform) could help place the findings in perspective.

4) Inclusion of seizure data for individual β1 and β4 spectrin knockout animals will help to interpret results of the double knockout. If you already have these data, it would be appropriate to add them. Otherwise, you might consider to remove this data point.

---

## [Author Response]

Essential revisions:The reviewers also found that there are several points that can be improved by including additional discussion.1) The discussion of localization of β1 spectrin and AnkyrinR in a subset of neurons should note that both of these proteins are present predominantly in the cell body, both in wildtype and β4 spectrin null neurons. While some β1 staining can be detected in the β4 knockouts, this is minimal in comparison to the cell body.

We revised the text as follows: “Remarkably, in *Nes-Cre;Sptbn4^F/F^* mice we found increased levels of AIS β1 spectrin in the majority of β1-high neurons (where somatic expression remained high; Figure 2D, arrow, E), but not at the AIS of β1-low neurons (Figure 2D, arrowheads, E).”

2) The Introduction is a single paragraph, and could be expanded.

We now include another introductory paragraph at the very beginning of the Introduction as follows:

“Clustered ion channels at axon initial segments (AIS) and nodes of Ranvier are essential for proper nervous system function. […] Using conditional knockout mice, we previously showed that loss of β4 spectrin from…”

3) The specialized function of β4 spectrin in recruiting CamKinase2 (missing in other β spectrins) should be mentioned. Similarly, discussion of specialized roles of giant AnkyrinG (ex-interactions with GABARAP, Nudel, EB proteins, and as a signaling platform) could help place the findings in perspective.

We revised the Discussion to include the suggested material as follows:

“β spectrins can also serve as a signaling platform. In cardiomyocytes, β4 spectrin regulates membrane excitability through coordinating CAMK2-mediated modification to Nav 1.5 channels for proper subcellular localization (Hund et al., 2010). […] For example, β4 spectrin-mediated signaling may influence downstream transcriptional regulation for axon regeneration after spectrin proteolysis by calpains.”

“AnkG’s giant exon participates in interactions with β4 spectrin (Jenkins et al., 2015), EB1/3 (Leterrier et al., 2011) and NDEL1 (Kuijpers et al., 2016). […] Giant AnkG also interacts with GABARAP to stabilize GABA receptors in the somatodendritic domain of neurons (Nelson et al., 2018).”

4) Inclusion of seizure data for individual β1 and β4 spectrin knockout animals will help to interpret results of the double knockout. If you already have these data, it would be appropriate to add them. Otherwise, you might consider to remove this data point.

We agree with the reviewers’ comment that seizure data for individual β1 and β4 conditional knockout mice would be very valuable. However, we did not perform EEG recordings from β1 conditional knockout mice since they had no phenotype whatsoever and brain morphology was completely normal. We did perform EEG recordings from 3 pairs of 6 month-old β4 conditional knockout mice (and control littermates). In these experiments we observed abnormal spike activity in *both* control and β4 conditional knockout mice, but seizures were not detected. In consultation with Dr. Jeffrey Noebels, who is an expert on mouse EEG recordings and who did this work, we decided not to include the β4 conditional knockout mice and only include the β1/β4 conditional knockout mice. We do not know how to explain the abnormal spike activity observed in the control mice. We concluded that new cohorts of conditional knockout and control mice would need to be prepared and the experiments performed again. This would require at least an additional 6 months. Because of the strong seizure phenotype in the β1/β4 conditional knockout mice, we strongly prefer to include these data found in Figure 4A-D.